# CVD Synthesis of Solid, Hollow, and Nitrogen-Doped Hollow Carbon Spheres from Polypropylene Waste Materials

**Pranav K. Tripathi** [†] [ID], **Shane Durbach** [‡] [ID] and **Neil J. Coville** * [ID]

DST-NRF Centre of Excellence in Strong Materials and the Molecular Sciences Institute, School of Chemistry, University of the Witwatersrand, Johannesburg 2050, South Africa; ptripathi2@gmail.com (P.K.T.); Shane.Durbach@twu.ca (S.D.)

* Correspondence: neil.coville@wits.ac.za
† Present Address: Indian Chemical Council, Sir Vithaldas Chambers, 16 Mumbai Samachar Marg Mumbai—400 001, India.
‡ Present Address: Department of Chemistry, Trinity Western University, 7600 Glover Road, Langley, BC V2Y 1Y1, Canada.

**Abstract:** Plastic waste leaves a serious environmental footprint on the planet and it is imperative to reduce this. Consequently, recycling has been regarded as an important approach in providing one solution to this problem. In this study, we enhanced the value of polypropylene (PP) plastic waste by using it as a hydrocarbon source to synthesize a variety of spherical carbon nanomaterials. Here, a CVD method was used to decompose the PP initially into a hydrocarbon gas (propylene). Thereafter, PP was employed to synthesize solid carbon spheres (SCSs), hollow carbon spheres (HCSs), and nitrogen-doped hollow carbon spheres (NHCSs). The latter two were made using a silica template while the N-doping was achieved by the addition of melamine to PP. Yields obtained were between 12–20%. The SCSs (d = 800 nm to 1200 nm), HCSs (id = 985 nm; shell width = 35 nm), and NHCSs (id = ca. 1000 nm; shell width = 40 nm) were all characterized by TEM, SEM, TGA, laser Raman spectroscopy, and XPS.

**Keywords:** hollow carbon spheres; nitrogen-doped hollow carbon spheres; carbon spheres; plastic waste; silica spheres

## 1. Introduction

The disposal of plastic waste material ranging from the nano- to the macro-scale has become a serious contemporary environmental issue [1–3]. This arises partly from the sheer volume of plastics produced (over $100 \times 10^9$ kg per annum [2]) and the lack of a 'cradle to grave' approach to the synthesis, manufacture, use, and disposal of plastics [4]. Many approaches have been proposed to address this issue, including the recent launching of the Alliance to End Plastic Waste (AEPW; www.endplasticwaste.org) by numerous plastics manufacturing companies to address the issue of plastic pollution of the environment.

Approaches to the disposal of plastic waste vary and include: 1) Generating polymers that will decompose with time via built-in chemical procedures [5,6], 2) The development of enzymes/bacteria that will decompose polymers [7], 3) Landfill options, and 4) Thermal decomposition by combustion [8]. In this latter option, the possibility exists to convert these waste plastics into high-value materials that could be re-used by society [9,10]. For example, plastic waste can be converted into spherical carbons and these spherical carbons can be solid or hollow and can be made with a range of sizes and porosities. These spherical carbons have been used in areas such as energy storage (batteries and

capacitors), adsorption, catalysis, drug delivery etc. [11–13]. This paper provides an example of this, describing how a common waste polymer, that is, polypropylene (PP) was converted into higher value spherical carbon nanomaterials.

PP is a common polymer found in society and is used in making films, fibers, tapes, etc. for use in the packaging of consumer products, in the automotive industry, and in textiles [14]. PP has also been used in more sophisticated items, such as centrifuge tubes—the material that was used in this study. Previous studies to convert waste polymers into high-grade chemicals have been reported. For example, the conversion of polystyrene and polyethylene into carbon nanotubes (CNTs), carbon nanofibers (CNFs) and carbon spheres (CSs) has been well documented [15–25]. Less work has been reported on the use of PP in similar studies.

Thermal studies on the conversion of PP to CNTs/CNFs have been successful, but the process required the use of a catalyst [24–26]. Attempts to produce spherical PP derived materials have been less successful. For example, the reaction of PP with Co(acetate)$_2$ as a catalyst produced a mixture of hollow carbon spheres (HCSs) and solid carbon spheres (SCSs) [25], while thermal plasma treatment of PP gave a mixture of poorly formed SCSs and CNTs [27].

The synthesis of SCSs [28,29] and HCSs [30,31] from hydrocarbons is well documented and their synthesis does not require the use of a catalyst. Thus, a simple process for the conversion of PP waste to produce a pure carbon material with a morphology dependent on the reaction conditions and choice of chemical reagents should be straightforward. Herein, we report on the use of used PP centrifuge tubes as a carbon source to make SCSs, HCSs, and nitrogen-doped HCSs (NHCSs).

## 2. Experimental Section

### 2.1. Chemicals

TEOS (98%, Aldrich), NH$_3$ (25% Fluka), isopropanol (Merck 99%) and deionized water were used as reagents for the synthesis of silica spheres. HF (40% Associated Chemicals) was used for silica removal. Monodisperse silica spheres with a diameter of 1000–1200 nm were synthesized according to a modified Stöber procedure [32,33].

### 2.2. Synthesis of SCSs

A similar procedure as used in [26] to make carbon materials from PP in a steel reactor was used in this study to make carbon materials in a quartz reactor. The same waste polypropylene (PP) centrifuge tubes were used as the hydrocarbon source. The PP tubes were washed, dried, and chopped into 0.5 cm × 8 cm pieces prior to use. The quartz tube reactor was placed in a two-stage CVD reactor, that is, a system containing two ovens that could be heated at two different temperatures (Supplementary Figure S1). The PP pieces were placed in the first stage of the reactor. The procedure used is provided in the Supplementary Materials (Figure 1).

The SCSs were obtained by the decomposition and vaporization of solid PP. In a typical synthesis, 2.0 g of used PP centrifuge tubes was placed in the quartz tube in the first stage of the CVD reactor while the temperature of the second stage reactor was increased from room temperature to 900 °C with a ramping rate of 10 °C·min$^{-1}$ under a flow of 100 sccm argon gas. Once the temperature of the second stage reactor was reached, the temperature of the first stage of the CVD furnace was ramped to 500 °C. The synthesis was maintained for 1 h followed by the natural cooling of the reactor. The SCSs were then collected from the quartz reactor.

### 2.3. Synthesis of HCSs

In a typical synthesis, 1.0 g of silica Stöber spheres was placed in a quartz boat in the second stage of the CVD reactor. Used PP centrifuge tubes (ca. 2 g), contained in a quartz boat were placed in the first stage of the CVD reactor. The temperature of the second stage reactor was increased from room temperature to 900 °C at a ramping rate 10 °C·min$^{-1}$ under a flow of 100 sccm argon gas.

Once the temperature of the second stage was reached, the temperature of the first stage of the CVD furnace was ramped to 500 °C. The propylene gas generated from the PP was deposited on the Stöber spheres. The reaction was maintained for 1 h and then the reactor was allowed to cool. This gave the silica/carbon material. HCSs were obtained by the removal of the silica template, with 10 mL of 10% HF solution for 24 h. Thereafter the HCSs were filtered and thoroughly washed with distilled water until a neutral pH of the washings was achieved.

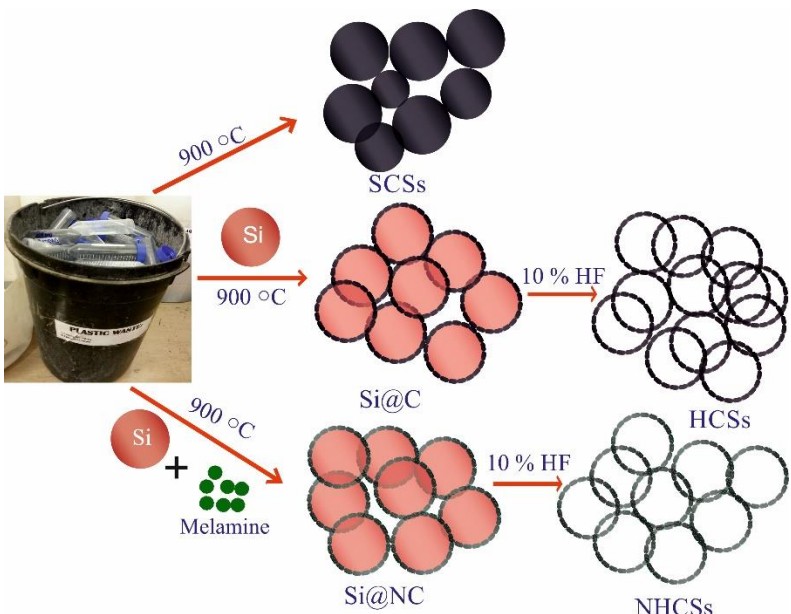

**Figure 1.** A graphical representation of solid carbon spheres (SCSs), hollow carbon spheres (HCSs), and nitrogen-doped hollow carbon spheres (NHCSs) synthesized from waste polypropylene (PP) material.

*2.4. Nitrogen-Doped HCSs*

The synthesis of the nitrogen-doped hollow carbon spheres was accomplished by the modification of the synthesis of the HCS material. The synthesis was performed by mixing melamine (1.0 g) with the PP centrifuge tubes (4–5 pieces; ca. 2 g), and this mixture was placed in the first stage of the reactor. Silica spheres were placed in the second stage of the reactor. All other conditions were the same as those used to synthesize the HCSs.

*2.5. Characterization*

The structural morphologies of these carbons were characterized by transmission electron microscopy (TEM, FEI Tecnai T12; Thermo Fisher Scientific, Hillsboro, Oregon, USA), accelerated at a voltage of 120 kV and scanning electron microscopy (SEM, FEI Nova Nanolab 600; Thermo Fisher Scientific, Hillsboro, Oregon, USA), accelerated at a voltage of 30 kV. Samples for TEM measurements were prepared by dropping a methanolic suspension of the various types of spheres onto a carbon film supported on a copper grid. The samples for SEM measurements were prepared by placing a carbon tape on top of a specimen holder and a small, representative amount of fine powder of the various types of spheres was dusted on the tape. Raman spectral measurements were made with the 514.5 nm line of a Lexel Model 95 SHG argon ion laser and a Horiba LabRAM HR Raman spectrometer (Horiba, Kyoto, Japan) with an Olympus BX41 microscope attachment. The laser was directed onto the sample with a 100× objective and the laser power at the sample was 0.4 mW. The beam spot size was a square of 10 micron × 10 micron, achieved by rastering the laser beam over a square with a DuoScan attachment. The backscattered light was dispersed via a 600-lines/mm grating onto a liquid nitrogen cooled CCD detector. The data acquisition software was LabSpec v5. The thermogravimetric Analysis (TGA) of the carbons was performed using a Perkin Elmer STA 6000 (Perkin Elmer, Waltham,

Massachusetts, USA), under an airflow of 20 mL·min$^{-1}$ with a ramping temperature 10 °C·min$^{-1}$ in the temperature range from 35 to 900 °C. The sample weight of carbons used for the TGA analysis was in the range 7.0 to 9.0 mg. The XPS analysis was performed on a Thermo ESCAlab 250Xi (Thermo Fisher Scientific, Hillsboro, Oregon, USA) using monochromatic X-rays produced from Al Kα radiation (1486.7eV electrons), and the experiments were performed at a pressure of 10$^{-11}$ bar. All XPS analyses were conducted at NMISA, CSIR, Pretoria.

## 3. Results and Discussion

Gaseous and liquid carbon materials can readily be used to make HCSs in a one-stage CVD reactor [29,34,35]. However, the use of solid carbon materials requires the conversion of the solid material into gaseous products in a preliminary step prior to decomposition of the carbon source. Thus, the conversion of the solid plastic waste investigated in this study, that is, used polypropylene centrifuge tubes, required a two-stage CVD furnace for the synthesis of carbon materials.

Polypropylene can readily be decomposed cleanly to propylene at T > 400 °C under inert gas conditions and thus a temperature of 500 °C was chosen for the first stage CVD furnace to make carbon spheres [26]. The temperature in the second stage of the reactor was initially chosen to range from 800 °C to 1000 °C to decompose the propylene. From this set of data, 900 °C was chosen for the study as this gave the SCSs a good yield and with diameters predominantly between 800 nm to 1200 nm (Supplementary Figure S2). The SEM images (Figure 2a) and TEM images (Figure 3a,b) of the SCSs, synthesized at 900 °C, show the expected morphology for the SCSs—a spherical shape with a smooth surface.

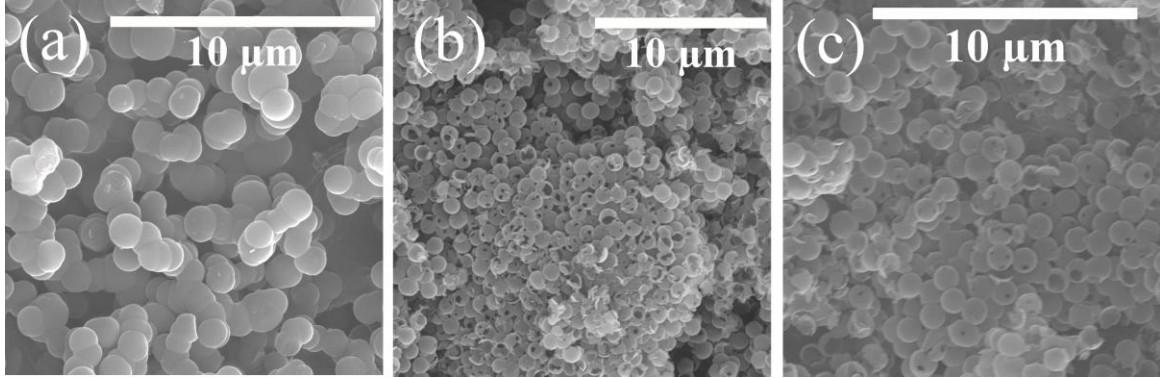

**Figure 2.** SEM images of the (**a**) SCSs, (**b**) HCSs and (**c**) NHCSs (900 °C, 1 h reaction).

The HCSs were made by a classical procedure [30,31], but using PP as the carbon source. The propylene produced from the PP readily decomposed on the SiO$_2$ spheres at 900 °C. The initial silica spheres had a spherical shape and a size of ca. 1100 nm; after heating at 900 °C the spheres, as expected, shrank (ca. 1000 nm). The carbon was then passed over the SiO$_2$ template to give Si/C spheres. The SiO$_2$ template was then removed by HF to give the HCSs. SEM and TEM images confirmed that the HCSs were hollow and had a diameter close to 1000 nm and a shell width of ca. 35 nm. (Supplementary Figure S2; Figure 2b, and Figure 3c,d). The SEM image showed that some of the HCSs had holes in the surface due to insufficient coverage of the template by the carbon. Longer reaction times and higher temperatures gave thicker shells.

The NHCSs materials were made by mixing melamine mixed with the solid plastic waste and by then heating the mixture [36]. The melting point of melamine (345 °C) [37] is similar to the *decomposition* temperature of PP (which melts at 130 °C). The removal of the SiO$_2$ template produced NHCSs material with a similar diameter (1000 nm) and a slightly thicker shell (40 nm) when compared to the HCSs material (Supplementary Figure S2; Figure 3e,f). This was due to the extra amount of reactants that were passed over the template (i.e., 1.0 g melamine and ca. 2 g PP), and was also mirrored by the lack of broken spheres seen in the SEM images of the NHCSs (Figure 2c). High magnification images

indicated that the HCSs and NHCSs sphere surfaces were rough (Figure 4). This was unexpected as most studies have shown that the surface of hollow carbons is relatively smooth [30,31]. The surface areas were measured and the SCSs had a low surface area as expected for CVD synthesized SCSs (4 m$^2$g$^{-1)}$), and these values increased to 25.7 m$^2$g$^{-1}$ (HCSs) and 42.1 m$^2$g$^{-1}$ (NHCSs) for the templated materials. These values are low, implying poor porosity. This will most likely have implications for the reactivity of the shells if these spheres are used in stabilizing metal catalysts on their carbon surfaces.

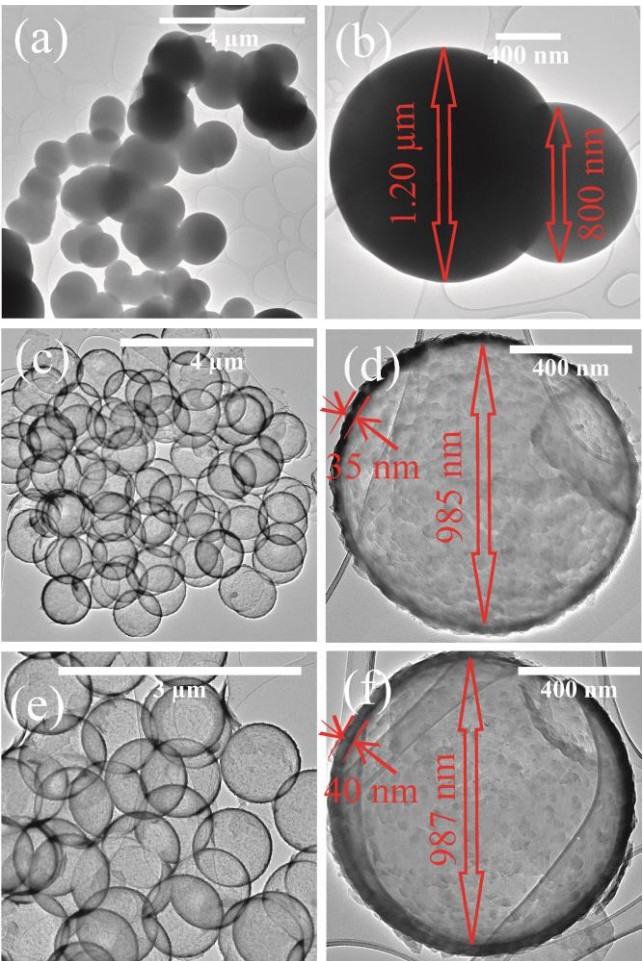

**Figure 3.** SEM images of (**a,b**) SCSs, (**c,d**) HCSs and (**e,f**) NHCSs (900 °C, 1 h reaction).

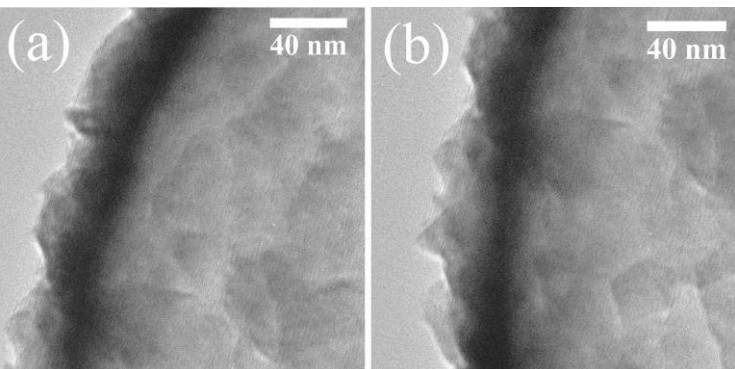

**Figure 4.** High magnification TEM images of the surface of (**a**) HCSs and (**b**) NHCSs.

TGA data were recorded for the PP and PP silica coated materials (not shown), and the silica residues determined from the carbon oxidation reaction in the TGA provided for a measure of the

carbon yields (ca. 18% for the HCS and ca. 12% for the NHCS materials). The TGA data for the SCSs, HCSs, and NHCSs materials are presented in Figure 5. The data for the SCSs show the expected loss of graphitic-like carbon with the derivative peak maximum between 600 °C and 750 °C, with no residual material remaining after the carbon oxidation. Interestingly, the HCSs show two derivative maxima, one at ca. 380 °C and the other at ca. 680 °C. The peak at 380 °C is suggestive of the loss of amorphous (or polymeric) carbon while the weight loss at 680 °C is consistent with loss of graphitic type carbon. The broader derivative peak observed for the solid spheres is expected. However, doping with N normally leads to a decrease in carbon stability that is not observed here. This could relate to the amorphous/polymeric material that was removed from the HCSs that led to easier oxidation of the remaining carbon in this material.

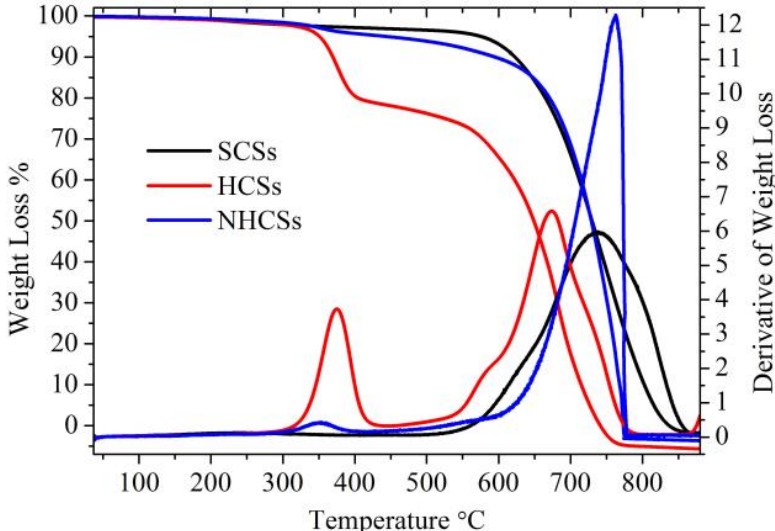

**Figure 5.** TGA and derivative curves for the SCSs, HCSs, and NHCSs.

Raman data were also recorded on the three spherical carbon nanomaterials (Figure 6). The typical carbon D and G bands were observed at ca. 1280 cm$^{-1}$ and ca. 1600 cm$^{-1}$ that indicate the amorphous/graphitic (and sp$^2$/sp$^3$) content of the carbon [38]. The G band position was invariant in the three samples while the D band showed an increase in peak position for the HCS and NHCS samples. The $I_D/I_G$ ratios for these bands gave values of 0.51 (SCSs), 0.50 (HCSs) and 0.54 (NHCSs) showing good aromatic content. The addition of N to the HCS indicated a modest increase in disorder, as expected for N-doped carbons. These values can be compared to HCSs (and NHCSs) made from traditional carbon sources such as acetylene and acetonitrile where more disordered graphitic structures were observed ($I_D/I_G$ > 1) [39,40]

XPS spectra were recorded for the three samples (Figure 7a; Supplementary Figures S3–S5 and Tables S1–S3). In all cases C and O were observed and some residual F (1%–2%) was noted in the NHCS and HCS materials. The residual F was readily removed by further washing with water. The C and O content of the SCS and HCS is given in Tables S2 and S3, and are typical of these types of materials. The N content was found to be 3.8%—again a common dopant loading found for NHCSs. Deconvolution of the N peak indicated four types of N atoms [41–43]: pyridinic N (21 %; 397,90 eV), pyrollic N (17%; 399.0 eV), graphitic N (45%; 400.82 eV), and N-O (17 %, 402.9 eV). These types of N atoms are found in varying ratios in N-doped graphitic carbon materials, and their ratios were determined by the nitrogen source, the carbon source, and the reaction temperature.

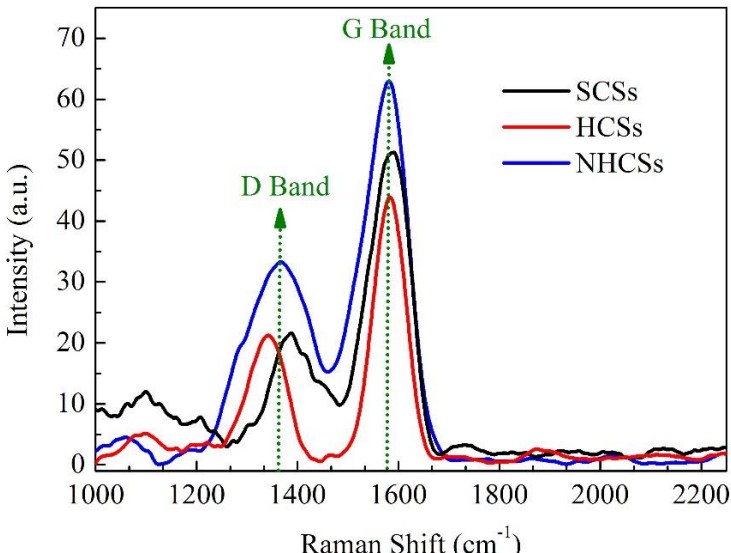

**Figure 6.** Raman spectra for the SCSs, HCSs and NHCSs.

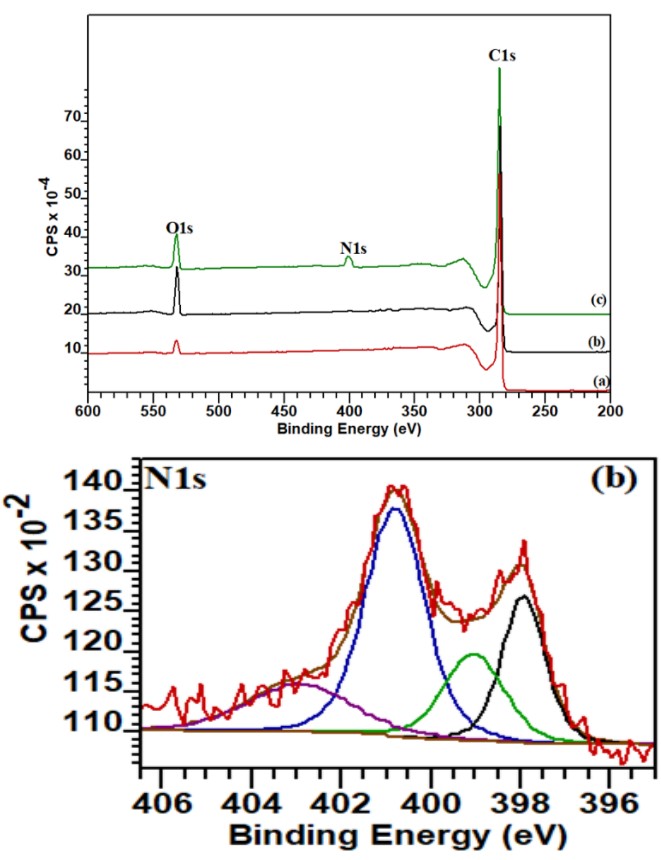

**Figure 7.** XPS spectra of (**a**) SCSs, HCSs, and NHCSs and (**b**) nitrogen spectrum for the NHCSs.

## 4. Conclusions

A facile method was developed to make spherical carbon materials from waste polypropylene centrifuge tubes. The CVD method entails the use of a two-stage reactor to convert the polypropylene into propylene, and finally into carbon spheres (CSs). The methodology used is similar to that used to make similar carbons (solid carbon spheres, hollow carbon spheres, N-doped hollow carbon spheres) from gaseous and liquid hydrocarbon sources. The CSs were characterized by standard procedures (TEM, SEM, XPS, TGA, Raman spectroscopy). The SEM and TEM data revealed that the carbons

had diameters of about 1000 nm while Raman data revealed that the material had good graphitic characteristics ($I_D/I_G < 1$)). The carbons are pure materials, contain no metal catalyst residues (as found in CNTs), and potentially can be used in many outlets where CSs are commonly used (batteries, fuel cells, capacitors, and as catalyst supports). A simple procedure is thus available to convert PP waste into higher value materials as an alternative to their placement in landfill sites.

**Supplementary Materials:** The following are available online at http://www.mdpi.com/2076-3417/9/12/2451/s1, Figure S1: (A) Schematic of the apparatus (modified form from ref [21]), (B) Schematic view of the two stage based CVD furnace used for the synthesis of CSs from polypropylene (a) SCSs, (b) Si@CSs and (c) Si@NCSs. Figure S2: Diameter distribution of (a) SCSs, (b) HCSs and (c) and NHCSs. Figure S3: High resolution XPS data for (a) C1s and (b) O1s XPS spectra of HCS sample. Figure S4: High resolution XPS data for (a) C1s, (b) N1s and (c) O1s XPS spectra of NHCS sample. Figure S5: High resolution XPS data for (a) C1s and (b) O1s XPS spectra of SCS sample. Table S1: Summary of XPS data. Table S2: XPS component peak positions. Table S3: Summary of % concentration of all bonds in the samples.

**Author Contributions:** Conceptualization, P.T.; methodology, P.T., N.C.; formal analysis, P.T., N.C., S.D.; writing—original draft preparation, P.T., N.C.; writing—review and editing, P.T., N.C., S.D.; supervision, S.D., N.C.; funding acquisition, N.C.

**Acknowledgments:** This research work was supported by the DST-NRF Centre of Excellence in Strong Materials (CoE-SM) at the University of the Witwatersrand. Authors are thankful to Boitumelo Matsoso for assistance with collecting the Raman spectroscopy data and for analysis of the XPS data. The authors are also thankful to the MMU, University of the Witwatersrand for the use of the TEM and SEM characterization facilities.

**Conflicts of Interest:** The authors declare no conflicts of interest.

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
