# Peer review of "CVD Synthesis of Solid, Hollow, and Nitrogen-Doped Hollow Carbon Spheres from Polypropylene Waste Materials"

_applsci, doi:10.3390/app9122451_

Round 1

Reviewer 1 Report

Dear Authors,

Congratulations on the experimental results. These are some minor comments that you can consider including in your manuscript.

Line 15-17 in the abstract. This sentence is too long and need to be broken down for easier read.

Line 27. The unit Mt is Mega Tonnes? Do spell it out.

Try to include "alternative to alchohol" or "pollution" in your introduction.

Line 29. Not sure if it is right to cite web-links in this format.

Line 47. Include the full chemical formula for acetate.

Line 93 could include the rough number of PP and melamine strips that make up the grams.

Figure 6. Maybe can smooth the graph and see if there is a shift? This is optional.

It will be great if a real picture and/or a schematic of the setup could be included.

Good luck

Author Response

We thank the referee for the kind comments. Response to comments follow:

 Attached a copy of paper showing changes made

Line 15-17 in the abstract. This sentence is too long and need to be broken down for easier read.

Thanks and done Thereafter, PP was employed to synthesize solid carbon spheres (SCSs), hollow carbon spheres (HCSs) and nitrogen doped hollow carbon spheres (NHCSs). The latter two were made using a silica template while the N-doping was achieved by the addition of melamine to PP.

Line 27. The unit Mt is Mega Tonnes? Do spell it out.

Change made and decided to use the more common unit of kg. Data was as quoted in the reference. 1 Mt = 1000000000 kg. so replaced with 109 kg

Try to include "alternative to alchohol" or "pollution" in your introduction.

I think this is what was being requested; lines 28/31:  ………use and disposal of plastics [4]. Many approaches have been proposed to address this issue, including the recent launching of the Alliance to End Plastic Waste (AEPW; www.endplasticwaste.org) by numerous plastics manufacturing companies to address the issue of plastic pollution in the environment.

 Line 29. Not sure if it is right to cite web-links in this format.

Not sure either. Editor to assist? It can be removed as well.

Line 47. Include the full chemical formula for acetate.

Done.

Line 93 could include the rough number of PP and melamine strips that make up the grams.

Added to text (4-5 strips depending on mass); melamine a powder.

Figure 6. Maybe can smooth the graph and see if there is a shift? This is optional.

Thanks and now done. Comment also made in text: The G band position was invariant in the three samples while the D band showed an increase in peak position for the HCS and NHCS samples.

It will be great if a real picture and/or a schematic of the setup could be included.

We have added this to the Supplementary material. It is taken from an earlier paper. Added to Fig S1

Reviewer 2 Report

The paper is interesting, especially because the authors have succeeded in the obtaining of an advanced material from plastic waste, also the preparation method is quite simple and catalyst-free. However, the paper has severe lacks and needs major revisions to be acceptable for publication.

Firstly, some minor correction should be made:

1.      The authors should replace Figure 1 to section 2.” Experimental” since the figure indicates the synthesis procedure and so, it is not part of the Introduction section.

2.      The introduction must give more information about the possible applications for this kind of materials and the advantages of the obtained shape.

3.      Authors must include some other references about carbon materials obtained from plastic wastes.

4.      Authors should improve de discussion of their materials comparing their results with other authors, for example, it is very interesting the low ratio ID/IG which indicates a high degree of graphitization. The comparison with other bibliography data would be very interesting and also, it would reinforce the novelty of the work.

5.      Authors should improve the conclusions.

Beside, mayor revision about:

6.      I highly recommend measuring the surface area of these materials, taking into account that they manifest roughness surface, and especially, because of the authors propose the materials as candidates for energy applications or as catalysts, where the BET area is one of the key factors.

7.      It would be very interesting to obtain the elemental analysis of these materials, and discuss the results versus the XPS data, since XPS analyses only the surface of the material, not the bulk.

8.      Finally, the most important point is that all XPS spectra are apparently wrongly interpreted. Authors have to be very careful with the assignment.

Firstly, spectra from Fig. S3 show the curve-fitting above the experimental data, it would mean that the theoretical quantity is higher than the experimental one which has no physical sense, moreover, authors have deconvoluted the spectra S3 a) b) d) only with one peak but it is evident that there are more contributions. Something similar happens with c) e) and f), c) and e) have more contributions with maybe lower FWHM.

Also, N spectra (Fig. 7) is completely wrong fitted, the fitting curve (red) has to be exactly on the experimental data line, it is also obvious the lack of another peak, and apparently, the graphitic and pyrrolic peaks are not well assigned.

Here the authors can find some reference to help them with the interpretation:

Doi: 10.1016/j.colsurfa.2017.02.021

        10.1007/s12678-018-0478-y

        10.1039/C7TA08023A

        10.1016/j.carbon.2014.09.005

Author Response

Thanks for the comments that certainly add to the paper.

Attached a copy of the paper showing changes made

Firstly, some minor correction should be made:

1.      The authors should replace Figure 1 to section 2.” Experimental” since the figure indicates the synthesis procedure and so, it is not part of the Introduction section.

Noted and done

2.      The introduction must give more information about the possible applications for this kind of materials and the advantages of the obtained shape.

This has been done. Three review articles that relate to applications have been added; plus two sentences on the applications. For example, plastic waste can be converted into spherical carbons and these spherical carbons can be solid or hollow and can be made with a range of sizes and porosities. These spherical carbons have been used in areas such as energy storage (batteries and capacitors), adsorption, catalysis, drug delivery etc. [11, 12, 13].

3.      Authors must include some other references about carbon materials obtained from plastic wastes.

We did have some references to making carbons from plastic waste (old refs 12-20) and have have added two further references on polystyrene.

4.      Authors should improve de discussion of their materials comparing their results with other authors, for example, it is very interesting the low ratio ID/IG which indicates a high degree of graphitization. The comparison with other bibliography data would be very interesting and also, it would reinforce the novelty of the work.

Thanks. We have added some references and a line on the ID/IG ratio issue. These values can be compared to HCSs (and NHCSs) made from traditional carbon sources such as acetylene and acetonitrile where more disordered graphitic structures were observed (ID/IG> 1) [39,40]

5.      Authors should improve the conclusions.

 The conclusion section has been expanded by adding the section below. A facile method has been developed to make spherical carbon materials from waste centrifuge polypropylene centrifuge tubes. The CVD method entails the use of a two stage reactor to convert the polypropylene into propylene and finally into the carbon spheres (CSs). The methodology used is similar to that used to make similar carbons (solid carbon spheres, hollow carbon spheres, N-doped hollow carbon spheres) from gaseous and liquid hydrocarbon sources. The CSs were characterized by standard procedures (TEM, SEM, XPS, TGA, Raman spectroscopy). The SEM and TEM data revealed that the carbons had diameters of about 1000 nm while Raman data revealed that the material had good graphitic characteristics (ID/IG < 1)).

Beside, mayor revision about:

6.      I highly recommend measuring the surface area of these materials, taking into account that they manifest roughness surface, and especially, because of the authors propose the materials as candidates for energy applications or as catalysts, where the BET area is one of the key factors.

Now added and comments made in text on values.The surface areas were measured and the SCSs had a low surface area as expected (4 m2g-1)and these values increased to 25.7 m2g-1(HCSs) and 42.1 m2g-1 (NHCSs) for the templated materials. These values are low, implying poor porosity.

7.      It would be very interesting to obtain the elemental analysis of these materials, and discuss the results versus the XPS data, since XPS analyses only the surface of the material, not the bulk.

Indeed, this is a good point and the surface values do not always reflect bulk values. In our case in-situ rather than post doping was used and so a homogeneous distribution of N is expected. .However, facilities are not readily available to us for these measurements to confirm this. 

8.      Finally, the most important point is that all XPS spectra are apparently wrongly interpreted. Authors have to be very careful with the assignment.

Firstly, spectra from Fig. S3 show the curve-fitting above the experimental data, it would mean that the theoretical quantity is higher than the experimental one which has no physical sense, moreover, authors have deconvoluted the spectra S3 a) b) d) only with one peak but it is evident that there are more contributions. Something similar happens with c) e) and f), c) and e) have more contributions with maybe lower FWHM.

Also, N spectra (Fig. 7) is completely wrong fitted, the fitting curve (red) has to be exactly on the experimental data line, it is also obvious the lack of another peak, and apparently, the graphitic and pyrrolic peaks are not well assigned.

Thanks for picking this up. The deconvolution of the N doped sample was done incorrectly and has now been corrected. S3 has also been corrected

More important I had not realised that the data had not been deconvoluted with CASA software, the software we normally use. I apologise for this error – we do lots of XPS work on these types of carbon materials and I had not checked the actual figures. We thank the referee for picking this up.

As a result, we have corrected all values and replaced all the XPS figures. 

Round 2

Reviewer 2 Report

Dear authors, 

Congratulations for the great work, after the second revision I highly recommend the present work for the publication in Applied Science journal. 

Authors have revised and modified the discussion according to the suggestions, and they have improved almost all the weak points previously commented, especially the XP Spectra, that are now perfectly interpreted. 

Now, the manuscript presents an appropriate quality to be published in Applied Science journal.

Just a couple of minor mistakes that I have now found:

Lines from 149 to 163 appear to be written in another typeface, also, the corresponding paragraph is not justified as well.

The line spacing is not the same throughout the text, just revise it.